# The Effect of Stretching Exercises Applied to Caregivers of Children with Development Disabilities on Musculoskeletal Muscle Mobility and Respiratory Function

**DOI:** 10.3390/ijerph21101361

**Published:** 2024-10-15

**Authors:** Amine Atac, Ebrar Atak

**Affiliations:** 1Department of Physiotherapy and Rehabilitation, Faculty of Health Sciences, Istanbul Gedik University, Istanbul 34876, Turkey; 2Department of Physiotherapy and Rehabilitation, Faculty of Health Sciences, Yalova University, Yalova 77100, Turkey; ebraratak@hotmail.com

**Keywords:** caregivers, health personnel, muscle mobility, musculoskeletal, myofascial theory, stretching exercise

## Abstract

We aimed to investigate the effect of stretching exercises applied to the hamstring, one of the posterior muscle chains, on musculoskeletal flexibility, chest mobility, and respiratory function. Proprioceptive neuromuscular facilitation and static stretching exercises were applied to 30 healthcare personnel caring for children with developmental delays using a crossover randomized study design. Posterior muscle chain mobility was assessed using the popliteal angle test (PAT) for the hamstring muscle, the mobility of the lumbar muscles was assessed using the Schober test (ST), and the mobility of the posterior chain muscles as a whole was assessed using the finger-to-floor distance test. Chest mobility was measured using chest circumference measurements and lung volumes were measured using the pulmonary function test (PFT). The results showed that stretching exercises applied to the hamstrings led to significant improvements in PAT, ST, and chest mobility in the direction of maximal expiration (*p* < 0.05), without being superior to each other. Ten males (33.3%) and twenty females (66.7%) who met the inclusion criteria were analyzed. The mean age of the participants was 26.6 ± 5.9 years, the mean height was 169.53 ± 8.67 cm, the mean weight was 65.26 ± 12.03 kg, and the mean body mass index was 22.58 ± 3 kg/m^2^. Chest inspiratory mechanics also showed a low positive correlation with posterior muscle mobility (r = 0.381; *p* = 0.038). There was no significant change in PAT. Within the framework of the myofascial theory, stretching exercises that can contribute positively to the musculoskeletal and respiratory system structures of healthcare professionals can be recommended and encouraged to healthcare professionals.

## 1. Introduction

Children with developmental disabilities can be dependent on rehabilitation health professionals and their families to meet their complex health needs [1]. Caregiver health studies are typically conducted on caregivers of children with cerebral palsy, autism, or mixed disabilities (which largely consist of children with these disabilities and children with Down syndrome) [2]. In the caregiver burden model, the body-hazardous tasks that caregivers perform every day have profound consequences on their health [3]. Parents or healthcare personnel caring for these children need to exhibit different abnormal postures, hold the child/patient for long periods of time, and have many other risk factors that may cause the development of musculoskeletal disorders [4]. It has been reported that the care demands of children with developmental delays negatively affect the caregivers’ musculoskeletal, cardiovascular, immune, and gastrointestinal systems, and somatic symptoms are seen in caregivers of children with disabilities [5,6]. Moreover, caring for a chronically ill child requires the investment of additional psychological, emotional, social, physical, and economic resources. Therefore, it poses a significant health problem for the parents and caregivers of children with disabilities who spend most of their lives in contact with such complex medical conditions [7]. In a screening conducted by Zaid et al. in 2022, it was found that the most common musculoskeletal problems in physiotherapists were in the neck and waist areas [8]. In this context, it is important to investigate interventions that may have a practical and immediate healing effect on improving the musculoskeletal system health of healthcare personnel working in rehabilitation and their caregivers.

The flexibility of the hamstring muscle is necessary to perform many activities of daily life effectively and efficiently. Despite this, hamstring muscle shortness is common in the general population [9]. This is attributed to the fact that this muscle is a biarticular posture muscle and tends to constantly shorten [10]. This situation causes many musculoskeletal system pathologies [11]. Considering the origin of the hamstring muscle, the shortening of the muscle causes the pelvis to move into a posterior pelvic tilt [12]. Following this, there is a decrease in lumbar lordase and an increase in thoracic kyphosis. This increase in kyphosis restricts chest expansion. Additionally, the length of the diaphragm and therefore its tension also changes. As a result, respiratory functions, respiratory muscle strength, and endurance are negatively affected [13]. The shortening of the hamstring muscle can cause various problems not only in the surrounding tissues but also in more distant areas [9]. Posterior chain muscles are examples of distant tissues that are affected by the shortness of this muscle [14]. Posterior chain muscles consist of the erector spinae, gluteus maximus, hamstring, gastrocnemius, soleus, and foot intrinsic muscles [15]. These muscles are the muscles that enable a person to maintain an upright posture against the force of gravity and surround the back of the body [16]. Tension in one of these chain muscles, including the hamstring muscle, leads to tension in the rest of the muscles in the chain. This is because the human body is designed according to a biological ‘tensegrity’. According to this idea, tissues are in equilibrium as a whole, as they are under compression and tension forces at the same time. Tension in one tissue causes tension in another tissue that is far away from that point [17]. If there is tightness and shortness in the hamstring muscle, tightness and problems can almost always be seen in the waist and shoulder muscles [14].

In the literature, there are studies explaining the relationship between hamstring muscles and respiratory parameters with the myofascial theory [9,16]. According to this theory, the human body is composed of fascia, a single tissue that functions as interconnected chains. Tension in one point of the fascia, which shows integrity, can result in tension or restriction in another part of the body [17]. In addition, some authors have suggested that approaches applied to the diaphragm may cause effects on distant tissues by transmitting myofascial tension [15]. In this context, researchers also took into account the limitations in the diaphragm when examining hamstring muscle flexibility. When we look at the future perspective, many studies can be planned within the scope of the myofascial theory, especially the function in the regions where it is dangerous to intervene, the function can be increased by intervening in a different part of the body. For this, different ideas and advanced study ideas with different designs are needed.

Stretching exercises are considered to have a very significant impact on joint flexibility, adding biomechanical precision to a person’s movement while providing the opportunity to perform at maximum strength throughout their range of motion [18]. Physical activity (PA) is defined as any body movement that is produced by skeletal muscles that requires energy expenditure, and exercise is one of the parameters of physical activity. People with low PA levels consume more medications and may struggle with a variety of health problems. While focusing on healthcare costs, encouraging healthcare professionals to engage in physical activity and exercise is very important for caregivers’ health [19]. Proprioceptive neuromuscular facilitation (PNF) stretching is known to be more effective than other stretching techniques as it increases both passive and active flexibility and improves the range of motion in the short term [20].

### Related Studies

In the literature, although health professionals, nurses, and physiotherapists examine the problems in musculoskeletal systems in their professional environments, there are few studies that reveal the effectiveness of exercise approaches that can benefit the musculoskeletal problems of caregivers. Especially in studies focusing on the musculoskeletal problems of caregivers, it is known that studies investigating the effect of a therapeutic intervention on any part of the respiratory system and musculoskeletal system, which are closely related to each other within the framework of the myofascial theory, are very few in the literature [9,15,16]. The hypothesis put forward in this study was made within the framework of the myofascial theory, which explains the relationship between the musculoskeletal system and the respiratory parameters and focuses on the fascia, which is a single tissue in the human body that functions as interconnected chains. This study aimed to investigate the effect of stretching exercises applied to the hamstring, one of the posterior muscle chains, on the respiratory functions, as well as the flexibility of the musculoskeletal system and the chest mobility of caregivers of children with developmental delays.

## 2. Materials and Methods

### 2.1. Study Design and Participants

The type of this study is a crossover randomized controlled trial. The sample of the study composed male and female healthcare workers who provide rehabilitation to children with developmental delays in rehabilitation centers between the ages of 18 and 40 years. Research data were collected from caregivers working at three different rehabilitation centers between 14 October 2023 and 15 May 2024. The study was carried out according to the Declaration of Helsinki and with the approval of the Istanbul Gedik University Scientific Research Ethics Committee (dated 24 August 2023, numbered E-56365223-050.02.04-2023.137548.174-548). The study’s clinical trial website registration number is NCT06466239.

The study included male and female participants aged 18–40 years who volunteered to participate in the study, participants without a history of infection or exacerbation in the 4 weeks preceding the study, participants who had not been involved in another clinical trial in the past 1 month prior to inclusion in the study, and both males and females with a knee flexion angle of 15 degrees or more in the hamstring muscle shortness test.

Participants with diagnosed conditions, such as vestibular disorders (vertigo, etc.), known balance disorders due to concussion in the past three months, musculoskeletal problems in the lower extremities (since the focus of our study was the hamstring muscle), those who had undergone lower extremity and thoracic surgery in the past year, and those with metabolic diseases, were excluded.

In the study, it was checked whether the caregivers of 38 children with developmental delays met the inclusion criteria. In the first stage, 30 caregivers who met the criteria were assigned to one of the proprioceptive neuromuscular facilitation (PNF) stretching group (PNFSGr) and the static passive stretching group (SSGr) using an online randomization method (www.randomiser.com). After the evaluations were made, the intervention procedure was applied and then the participants were evaluated again. A 1 day washout period was allowed to remove the effectiveness of the first application. One day later, the participants underwent a stretching procedure that was not applied in the first phase. The randomized controlled crossover study design is described in the flow diagram (Figure 1).

### 2.2. Measurement Instruments

Information such as gender, age, height, body mass index, complaints, comorbidities, discomforts experienced in the past 4 weeks, family history, medical history, presence of recurrent infections, and medications used by the caregivers were questioned with the demographic data form. Posterior muscle chain mobility was evaluated using the popliteal angle test (PAT) with a goniometric measurement for the hamstring muscle, the mobility of the lumbar muscles was evaluated using the Schober test (ST), and the finger-to-floor distance test (FFDT) was used to evaluate the mobility of the posterior chain muscles as a whole. In the PAT procedure, the caregiver was laid down on their back with the hip and knee joints of the side to be evaluated at 90° flexion and the contralateral side with the extremity in extension. The participant was told to hold the back surface of their leg with their hands and maintain the flexion angle. Meanwhile, the inclinometer was kept at the anterior midline of the tibia. The person was then asked to extend their knee as much as they could tolerate, and the measurement was taken at the last degree of movement [21]. During the Schober test, while the person was standing, the midpoint at the posterior superior level of the spina iliaca and 10 cm above this point were marked. The person was then asked to lean forward and the distance between the two marked points was measured again using a tape measure. The difference between the two measurements was recorded. To talk about normal lumbar mobility, a difference of 5 cm was determined as the reference value. It has been stated in the literature that this measurement has a high and moderate validity compared with radiography [16]. In the FFDT evaluation, the person standing on a block with a height of 15 cm from the ground was asked to lean forward and touch their toes without bending their knees. At the end point of the movement, the distance between the distal end of the extended fingers and the ground was measured. Values above the block were considered as minus, values below were considered as positive, and the measurement was repeated three times and the average value was recorded [9].

While evaluating chest mobility, a chest circumference measurement (CCM) was made by measuring the normal resting tidal volume, maximum inspiration, and maximum expiration time from the axillary, epigastric (xiphoid), and subcostal regions with a tape measure and recording it in centimeters. Respiratory functions were determined using a pulmonary function test (PFT). Using the COSMED microQuark spirometer device with a personalized anti-viral filter, measurements were made while the caregiver was in a sitting position and the lung volumes (FVC, FEV1, FEV1/FVC, PEF values) were recorded.

### 2.3. Procedures

In the PNF stretching application, the hold–relax technique for each muscle was applied to the caregiver as self-stretching with the concept of direct treatment. While the muscle to be stretched was in the standing position for the caregiver (the antagonist muscle for the pattern—the muscle causing limitation), it was first brought to the position where it could extend and at that point, the stretched muscle was contracted to a submaximal level (65% of the intensity determined according to the Stretch Intensity Scale) for 5 s (sec) as a way to initiate isometric contraction. Then, five relax commands were given to the caregiver for the tensed muscle. Then, the stretched muscle was stretched to the point it could reach and the muscle to be stretched was passively stretched for 20 s (one set) [22]. After a 5 s rest interval [20], the method was repeated and eight sets (4 min duration) were performed in one muscle [22]. For the hamstring muscle, the person positioned their lower extremity on a step with their knee straight, i.e., in extension, and leaned as far as they could with their hand on the toe of the foot with their back straight. Then, in this position, the isometric contraction of the hamsting muscle was made for 5 s by pressing the heel against the step. Then, after relaxing the muscle and waiting for 5 s, they extended their hand in the direction of the fingertip and waited for 20 s at the last point where the muscle could extend. In the static stretching application, the classical static stretching application was performed on the hamstring muscle for 30 s each, eight repetitions, 4 min for one leg, and 8 min for both legs in the posture position in PNF stretching.

### 2.4. Data Analysis

We used the a priori method with the help of the G-Power (Ver. 3.1.9.7) program to determine the sample size before conducting the study and during the design and planning phase. This method allows for the calculation of the sample size required to achieve the desired α level and power level (1-β) to estimate the effect size. An a priori analysis is an ideal method to test the hypothesis, as it controls for Type I and Type II errors. In the literature, a minimum of 28 participants were included in the study [23]. This study used an effect size of 0.71, a power of 95%, and a margin of error of 0.05.

In this study, the Shapiro–Wilk test was used to assess whether the continuous variables were normally distributed. Since the sample size was small, the analyses indicated that none of the variables were normally distributed. Therefore, we used non-parametric tests. When all the assumptions are met, non-parametric tests are less sensitive than their parametric counterparts, meaning that larger differences are needed for a non-parametric test to reject the null hypothesis. Non-parametric tests also tend to use less information than parametric ones. However, when the assumptions are not met (e.g., data are not normally distributed), non-parametric tests are the ideal choice.

The descriptive statistics for the variables are expressed as mean ± standard deviation and median (minimum–maximum). The descriptive statistics for the categorical variables are reported as n (%). To compare the differences between two independent groups when the variables are not normally distributed, we used the Mann–Whitney U test.

To compare two related samples, such as paired differences in repeated measurements (e.g., before-and-after treatment measures in this study), we used the Wilcoxon signed-rank test. The Spearman correlation test was employed for comparisons of two continuous variables that were not normally distributed.

In the analysis, *p* < 0.05 was considered significant. We used the common threshold of *p* < 0.05, which indicates that the data are likely to occur less than 5% of the time under the null hypothesis. When the *p*-value falls below the chosen alpha value, the result of the test is considered statistically significant.

The data were analyzed using the SPSS 22.0 program.

## 3. Results

Thirty-eight individuals participated in the study, and thirty individuals who met the inclusion criteria and had no missing data were included (Figure 1). Ten males (33.3%) and twenty females (66.7%) who met the inclusion criteria were analyzed. The mean age of the participants was 26.6 ± 5.9 years, the mean height was 169.53 ± 8.67 cm, the mean weight was 65.26 ± 12.03 kg, and the mean body mass index was 22.58 ± 3 kg/m^2^. The distribution of the descriptive characteristics of the caregivers is summarized in Table 1.

The changes in posterior muscle chain mobility before and after stretching are shown in Table 2. The changes in chest circumference measurements before and after stretching are shown in Table 3. For the PAT, ST, FFDT, and CCM evaluations performed before and after PNF stretching and static stretching applications, those who did not show significant changes (*p* > 0.05) and those who showed significant changes in negative and positive directions (*p* < 0.05) are shown in Table 2 and Table 3. In the PFT evaluation, in which the lung volumes were measured, there was no significant difference in the change of the values before and after both types of stretching (*p* > 0.05).

In the comparison of the differences in the measurements made before and after the applications, there was no superiority of the applications against each other (*p* > 0.05) (Table 4). In the correlational analysis of the measurement differences before and after the stretching applications, a positive, low level, and statistically significant relationship was found in the FFDT and subcostal region chest circumference measurement at maximal inspiration (SCCMI) variables (*p* < 0.05) (Table 5).

## 4. Discussion

The aim of this study was to investigate the effect of stretching exercises applied to the hamstring, one of the posterior muscle chains, on the musculoskeletal flexibility, chest mobility, and respiratory function of caregivers of children with developmental delays. The results showed that stretching exercises applied to the hamstring increased posterior muscle chain mobility and improved chest mobility in the expiratory direction. Chest inspiratory mechanics also showed a low positive correlation with posterior muscle mobility.

One of the hypotheses of this study was to reveal the effectiveness of stretching exercises in children with developmental delays, in which caregivers can have an immediate effect on increasing the posterior muscle chain and thoracic mobility and provide a positive effect on work performance. In a study conducted by Wadeson et al. in 2020 in eight male gas cylinder drivers and transporters, it was found that stretching exercises had an effect on mobility-related muscle activity in most of the muscle groups evaluated and positively affected the working performance of the employees [24]. This study also revealed that stretching exercises have a positive acute effect on muscle mobility, similar to Wadeson’s study, although the muscles applied to rehabilitation caregivers tend to increase muscle tension due to the caregiver burden. In the study by Reiner et al. in 19 participants who did not have any pathology in 2021, it was found that PNF stretching exercises applied alone to the gastrocnemius, which is in the posterior muscle chain muscles, contributed to mobility by increasing the normal joint range of motion of the ankle [25]. In this study, different stretching exercises were applied to the hamstring muscle, and it was investigated whether they would additionally affect chest mobility and respiratory functions. Brandão et al., in 2023, examined the acute effect of PNF stretching on the hamstring and gastrocnemius muscles, examining its effect on posterior muscle chain flexibility in 15 healthy participants [26]. Their study showed that the stretching protocol is effective in improving joint mobility but is not sufficient to elicit immediate mechanical changes in muscle and tendon stiffness. In this study, hamstring flexibility was measured by measuring the popliteal angle rather than the normal joint movement measurement, and an acute positive effect of stretching exercises on mobility was observed. In this study, three different tests were used to evaluate posterior muscle chain mobility. Except for FFDT, there was a significant improvement in the other two evaluations, PAT and the Schober test, after both stretching applications. The reason for not finding a significant difference in the FFDT evaluation may be due to the mechanism of the test being performed, rather than muscle mobility and flexibility in the pre-stretching evaluation, and the difference between the before and after evaluations after movement from the hip joint. According to the literature, PAT and the Schober test are considered reliable in evaluating posterior muscle chain mobility [27]. We can say that there are differences and similarities between the study of Brandão et al. and the protocols of this research, but the common result is that stretching exercises positively affect posterior muscle chain mobility and joint movements. In this context, we can encourage healthcare personnel with care burdens to apply static or PNF stretching trainings that they can apply instantly in their professional lives in order to protect the mobility health of their muscles, joints, and soft tissues.

In a study conducted in 2022, it was investigated whether PNF stretching exercises applied to the chest pectoral muscles of patients with COVID-19 with mild and moderate impairment for seven sessions had an effect on chest mobility and lung volumes [28]. Contrary to the results of this study, they observed improvements in the results of the pulmonary function tests that evaluate lung volumes. The reason for the difference in the results may be that, since this study looked at the acute effect of stretching exercises, instant stretching had an insufficient effect on increasing lung volumes. In addition, since the population of our study was caregivers who did not have respiratory pathology, such as COVID-19, the lack of a significant difference between the pre- and post-respiratory function test evaluations may be explained. While the study in the literature looked at the effect of stretching applied directly to the chest pectoral muscles for seven sessions and revealed positive effects on chest mobility, this study applied stretching to the hamstring, a muscle independent of breathing, and used the myofascial theory focusing on the fascia, a single tissue that functions as interconnected chains in the human body. It revealed that they had a significant improving effect on chest expiratory mobility in a supportive direction. Apart from this study, which showed insufficient results in affecting lung volumes due to the acute effect, we can recommend that future literature studies investigate whether stretching applications in the hamstring muscle over a period of weeks affect lung volumes. In a study by Mehta et al. conducted in 2015 in the elderly, PNF stretching was applied to the chest muscles along with a supervised exercise program for 1 week and chest mobility and respiratory functions were evaluated [29]. While they obtained a significant improvement in chest expansion, they found a change in lung volumes only in the FEV1/FVC value. In this study, unlike the aforementioned study, a significant improvement was seen only in chest expiration, and no acute effect of instantly applied PNF or a static stretching session was observed on any of the lung volumes. Additionally, in this study, a positive relationship was found between the subcostal circumference measurement at maximum inspiration and the FFDT evaluation, which evaluates posterior muscle chain mobility. The different results between these two studies may be due to the difference in the populations, the day and duration of the stretching sessions, and the muscle group in which the stretching was performed. Within the framework of the myofascial theory, the fascia is a whole and the technique performed in one part of the body can affect the tissues in another part of the body, with the hypothesis that the technique performed in one part of the body can affect the tissues in another part. Metha et al. performed stretching on the pectoral muscles for 1 week, while we performed one session of instantaneous stretching on the hamstring muscle and looked at the acute effect on chest mobility.

Csepregi et al. 2022 investigated the effects of classical breathing exercises on spine and chest mobility in female university students [30]. For 7 weeks, only classical breathing exercises were applied to one group and yoga and pilates programs were applied to the other two groups. They found that classical breathing exercises had a significant effect on chest and spine mobility in the chest circumference measurement, Schober test, and FFDT values, despite being applied alone. In 2015, Valenza et al., and in 2016, Álvarez et al., looked at the various effects of the stretching procedure applied to the diaphragm muscle from the posterior muscle chains to the hamstring and found a positive effect on posterior muscle chain mobility and flexibility [9,16]. In this study, we hypothesized that the fascia is a whole and that stretching the hamstring muscle, which is far from the thorax, may affect chest mobility, and we looked at the effects of acute one-time static and PNF stretching. In this study, we found that only mobility improvement improved expiration. A therapeutic intervention applied in the hamstring had a positive effect on thoracic mobility despite a single acute application. In the study by Csepregi et al., a 7-week therapeutic intervention involving the chest area using breathing exercises could affect spinal mobility. In these studies, the fact that therapeutic interventions applied in unrelated regions affected a region that was different from the applied region can be explained by the “Biotensegrity Model”, one of the fascial models [17,31]. In this model, which states that the fascia is a continuous piece of tissue that works in interconnected “chains” to create tension in the body, it is stated that when the fascia is stretched in one region, it may cause tension, restriction, and pain in another part of the body. In this study, the stretching of the hamstring from the posterior muscle chain may have caused tension in the fascia related to chest mobility, while in other studies, tension in the structures related to the thorax and chest may have caused elongation in the fascia of the posterior muscle chain and spinal structures, causing a significant effect to be observed.

### Implications, Limitations, and Future Perspectives

The hypothesis of this study was that a therapeutic intervention applied to the hamstring within the framework of the myofascial theory could affect mobility and respiratory parameters. As a result of some studies in the literature [32,33,34,35,36,37] and this study, our conclusion is that the fascia is a whole and therapeutic intervention in the musculoskeletal units at one point of the body that may affect the units at other points of the body and change the overall performance and functions. This study had some limitations. The first limitation is the lack of a control group with a placebo stretching application other than PNF and the static stretching applications in the study. Since this study was a crossover design, a placebo control group could have been created, although no gender-based comparison was made. Another limitation was that the stretching applications in the study design had an insufficient application time to change the lung volumes in the respiratory function test. In future studies, the effect of the stretching applications applied to the hamstring muscle regularly for weeks on the chest mobility and respiratory function test results, rather than the acute effect, can be examined in the caregivers of people with developmental disabilities. When we look at the future perspective, many studies can be planned within the scope of the myofascial theory, especially the function of the regions where it is dangerous to intervene, as the function can be increased by intervening in a different part of the body. For this, different ideas and advanced study ideas with different designs are needed.

## 5. Conclusions

In this study, it was observed that both PNF and static stretching had an improving effect on musculoskeletal mobility and also on chest expiratory mobility despite stretching the hamstring. In this context, since stretching exercises can acutely improve mobility in most muscle groups, regardless of region, and indirectly have a positive effect on work performance, we can recommend that caregivers gain the habit of stretching exercises that they can do daily in their working lives. In order to maintain or improve chest mobility, caregivers can benefit from self-applied static or PNF stretching to the hamstring, one of the posterior chain muscles, independently of the chest area, during their work lives and breaks. As a result of this study, easy-to-apply stretching exercises that can positively contribute to the musculoskeletal and respiratory system structures of healthcare professionals were recommended to healthcare professionals within the framework of myofascial theory.

## Figures and Tables

**Figure 1 ijerph-21-01361-f001:**
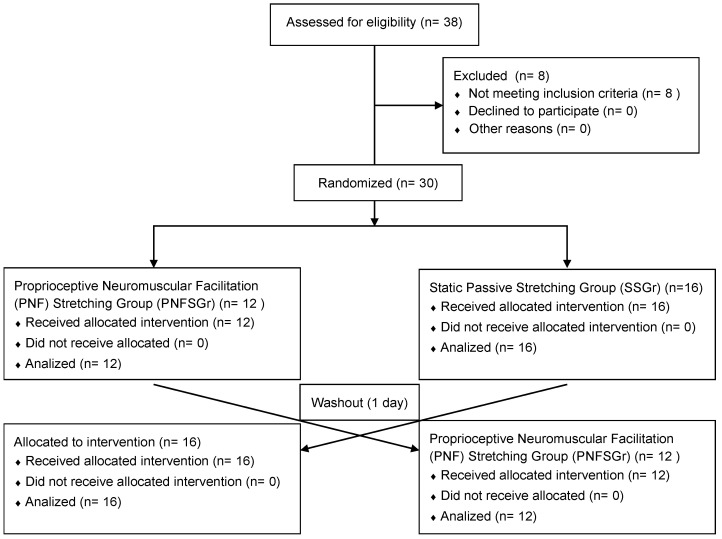
Study flow diagram.

**Table 1 ijerph-21-01361-t001:** Distribution of the descriptive characteristics of the caregivers.

	n = 30	%		
Gender				
Female	20	66.7		
Male	10	33.3		
		Min.–Max.	Mean ± SD	Median
Age (years)	30	21–40	26.6 ± 5.9	24
Height (cm)	30	156–188	169.53 ± 8.67	167
Weight (kg)	30	48–92	65.26 ± 12.03	63.5
BMI (kg/m^2^)	30	18.68–30	22.58 ± 3	22.65

n: number, %: percent, min: minimum, max: maximum, SD: standard deviation, cm: centimeter, kg: kilogram, BMI: body mass index, m^2^: square meter.

**Table 2 ijerph-21-01361-t002:** Changes in posterior muscle chain mobility before and after stretching applications.

	PNFSGr				SSGr			
Variables	Changing	n = 30	Z	*p*-Value	Changing	n = 30	Z	*p*-Value
PATright	Negative changing	24	−3.808	<0.001 *	Negative changing	25	−3.271	0.001
	Positive changing	4			Positive changing	3		
	Unchanging	2			Unchanging	2		
PATleft	Negative changing	25	−4.442	<0.001 *	Negative changing	29	−4.441	<0.001 *
	Positive changing	2			Positive changing	1		
	Unchanging	3			Unchanging	0		
Schober Test	Negative changing	3	−2.856	0.004	Negative changing	2	−3.93	<0.001 *
	Positive changing	17			Positive changing	23		
	Unchanging	10			Unchanging	5		
FFDT	Negative changing	17	−0.370	0.712	Negative changing	16	−0.476	0.634
	Positive changing	9			Positive changing	13		
	Unchanging	4			Unchanging	1		

Wilcoxon test, * *p* < 0.05, PNFSGr: proprioceptive neuromuscular facilitation (PNF) stretching group, SSGr: static passive stretching group, PAT: popliteal angle test, FFDT: finger-to-floor distance test. A negative change in the result value for PAT indicates increased hamstring flexibility and a positive change in the result value for the Schober test indicates increased posterior muscle chain mobility.

**Table 3 ijerph-21-01361-t003:** Changes in chest mobility before and after stretching applications.

	PNGSGr				SSGr			
Variables	Changing	n = 30	Z	*p*-Value	Changing	n = 30	Z	*p*-Value
ACCTV	Negative changing	5	−1.725	0.084	Negative changing	3	−1.633	0.102
	Positive changing	1			Positive changing	0		
	Unchanging	24			Unchanging	27		
ACCMI	Negative changing	9	−0.137	0.891	Negative changing	9	−0.961	0.337
	Positive changing	9			Positive changing	13		
	Unchanging	12			Unchanging	8		
ACCME	Negative changing	13	−2.215	0.027 *	Negative changing	17	−1.21	0.226
	Positive changing	5			Positive changing	9		
	Unchanging	12			Unchanging	4		
ECCTV	Negative changing	2	−0.447	0.655	Negative changing	7	−1.81	0.07
	Positive changing	3			Positive changing	2		
	Unchanging	25			Unchanging	21		
ECCMI	Negative changing	10	−0.268	0.788	Negative changing	14	−0.912	0.362
	Positive changing	12			Positive changing	6		
	Unchanging	8			Unchanging	10		
ECCME	Negative changing	13	−1.228	0.219	Negative changing	19	−3.231	0.001 *
	Positive changing	9			Positive changing	3		
	Unchanging	8			Unchanging	8		
SCCTV	Negative changing	1	−1.414	0.157	Negative changing	5	−0.551	0.582
	Positive changing	4			Positive changing	4		
	Unchanging	25			Unchanging	21		
SCCMI	Negative changing	10	−0.885	0.376	Negative changing	10	−0.55	0.583
	Positive changing	12			Positive changing	10		
	Unchanging	8			Unchanging	10		
SCCME	Negative changing	9	−0.205	0.837	Negative changing	13	−0.198	0.843
	Positive changing	9			Positive changing	8		
	Unchanging	12			Unchanging	9		

Wilcoxon test, * *p* < 0.05, PNFSGr: proprioceptive neuromuscular facilitation (PNF) stretching group, SSGr: static passive stretching group, ACCTV: axillar chest circumference in tidal volume, ACCMI: axillar chest circumference in maximal inspiration, ACCME: axillar chest circumference in maximal expiration, ECCTV: epigastric chest circumference in tidal volume, ECCMI: epigastric chest circumference in maximal inspiration, ECCME: epigastric chest circumference in maximal expiration, SCCTV: subcostal chest circumference in tidal volume, SCCMI: subcostal chest circumference in maximal inspiration, SCCME: subcostal chest circumference in maximal expiration. A negative change in the maximal expiration means increased expiratory mobility.

**Table 4 ijerph-21-01361-t004:** Comparison of the differences between before and after the stretching applications.

Variables	Stretching Type	n = 30	Mean ± SD	Median	U	*p*-Value
PATright	Static Stretching	30	6.06 ± 7.85	−5	424	0.699
	PNF Stretching	30	5.56 ± 5.1	−4.5		
PATleft	Static Stretching	30	4.8 ± 5	−5	438	0.858
	PNF Stretching	30	5.33 ± 4.05	−5		
Schober Test	Static Stretching	30	0.51 ± 0.63	0.5	385.5	0.318
	PNF Stretching	30	0.41 ± 0.71	0.5		
FFDT	Static Stretching	30	1.34 ± 6.01	−1	436	0.835
	PNF Stretching	30	0.22 ± 2.45	−1		
ACCTV	Static Stretching	30	0.13 ± 0.43	0	432	0.668
	PNF Stretching	30	0.4 ± 1.4	0		
ACCMI	Static Stretching	30	0.43 ± 2.22	0	401.5	0.461
	PNF Stretching	30	0.26 ± 1.14	0		
ACCME	Static Stretching	30	0.33 ± 1.32	−1	428.5	0.743
	PNF Stretching	30	0.5 ± 1.16	0		
ECCTV	Static Stretching	30	0.33 ± 1.06	0	366.5	0.095
	PNF Stretching	30	0.03 ± 0.41	0		
ECCMI	Static Stretching	30	0.46 ± 1.87	0	390.5	0.368
	PNF Stretching	30	0.13 ± 1.52	0		
ECCME	Static Stretching	30	1.03 ± 1.5	−1	348	0.123
	PNF Stretching	30	0.5 ± 2	0		
SCCTV	Static Stretching	30	0.1 ± 0.88	0	396.5	0.285
	PNF Stretching	30	0.23 ± 0.97	0		
SCCMI	Static Stretching	30	0.16 ± 1.68	0	433.5	0.803
	PNF Stretching	30	0.23 ± 1.4	0		
SCCME	Static Stretching	30	0.06 ± 1.31	0	407.5	0.515
	PNF Stretching	30	0.06 ± 1.31	0		

Mann–Whitney U test, *p* < 0.05, PNFSGr: proprioceptive neuromuscular facilitation (PNF) stretching group, SSGr: static passive stretching group, PAT: popliteal angle test, FFDT: finger-to-floor distance test, ACCTV: axillar chest circumference in tidal volume, ACCMI: axillar chest circumference in maximal inspiration, ACCME: axillar chest circumference in maximal expiration, ECCTV: epigastric chest circumference in tidal volume, ECCMI: epigastric chest circumference in maximal inspiration, ECCME: epigastric chest circumference in maximal expiration, SCCTV: subcostal chest circumference in tidal volume, SCCMI: subcostal chest circumference in maximal inspiration, SCCME: subcostal chest circumference in maximal expiration.

**Table 5 ijerph-21-01361-t005:** Correlational analysis of the stretching applications difference analysis.

Variables **		ACCTV	ACCMI	ACCME	ECCTV	ECCMI	ECCME	SCCTV	SCCMI	SCCME	ST	FFDT	PATright	PATleft
ACCTV	r	0.129												
	*p*	0.495												
ACCTV	r		−0.180											
	*p*		0.342											
ACCTV	r			−0.282										
	*p*			0.131										
ECCTV	r				0.318									
	*p*				0.087									
ECCTV	r					0.070								
	*p*					0.714								
ECCTV	r						−0.286							
	*p*						0.125							
SCCTV	r							−0.255						
	*p*							0.173						
SCCTV	r								0.381 *					
	*p*								0.038 *					
SCCTV	r									0.287				
	*p*									0.124				
ST	r										0.241			
	*p*										0.199			
FFDT	r											0.569 *		
	*p*											0.001 *		
PATright	r												0.052	
	*p*												0.786	
PATright	r													0.088
	*p*													0.643

Spearman’s correlation test, * *p* < 0.05, ** variables in the horizontal column represent the difference in PNFSGr and variables in the vertical column represent the difference before and after SSGr. ACCMI: axillar chest circumference in maximal inspiration, ACCME: axillar chest circumference in maximal expiration, ECCTV: epigastric chest circumference in tidal volume, ECCMI: epigastric chest circumference in maximal inspiration, ECCME: epigastric chest circumference in maximal expiration, SCCTV: subcostal chest circumference in tidal volume, SCCMI: subcostal chest circumference in maximal inspiration, SCCME: subcostal chest circumference in maximal expiration, ST: Schober test, PAT: popliteal angle test, FFDT: finger-to-floor distance test.

## Data Availability

The data that support the findings of this study will be made available by the corresponding author upon reasonable request. Data are not publicly available due to privacy and ethical concerns.

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
