# Peer review of "The Effect of Stretching Exercises Applied to Caregivers of Children with Development Disabilities on Musculoskeletal Muscle Mobility and Respiratory Function"

_ijerph, 2024, doi:10.3390/ijerph21101361_

Round 1

Reviewer 1 Report

Comments and Suggestions for Authors
The topic addressed is highly relevant, particularly considering the importance of the physical well-being of caregivers of children with developmental disabilities. The study focuses on a research area that has a direct impact on the quality of life of a specific and often overlooked population. The study design is well-conceived, utilizing a randomized crossover trial structure that ensures robustness in data collection and analysis. The use of specific and validated measures to assess musculoskeletal mobility and respiratory functions is an additional strength.

However, some improvements are necessary:
  1. It is recommended to introduce a "1.2 Related Studies" section within the initial part of the paper. This section could provide an overview of relevant previous research, highlighting how these studies have employed similar techniques or addressed analogous topics.
  2. The description of inclusion and exclusion criteria could be enriched with additional details to enhance the understanding of the study population's characteristics.
  3. Paragraph 2.4 (which I would rename "Statistical Data Evaluation") should be made more precise. The text mentions that a power analysis was conducted to determine the minimum number of participants, but it does not clearly specify the type of test used to calculate effect size and power. It would be helpful to clarify whether the analysis refers to a t-test, ANOVA, or another type of test. Additionally, while it is stated that the power analysis was performed using the G-Power program, the type of analysis conducted within G-Power (e.g., a priori, post hoc) is not specified, which could be clarified to make the information more precise. Furthermore, the text states that the data did not show a normal distribution, but it is not specified whether this applies to all variables or only some. It would be useful to indicate which variables were tested and which did not show a normal distribution. The text briefly describes the statistical tests used (Wilcoxon, Mann-Whitney, Spearman), but it might be helpful to specify in which situations or for which types of data these tests were applied. For example, it would be useful to clarify whether the Wilcoxon test was used to compare repeated measurements of the same group. The text indicates that a p-value of <0.05 was considered significant, but it might be beneficial to briefly explain the meaning of this threshold and how it is interpreted in the specific context of the study. Lastly, the results of the statistical analyses could be better contextualized, for example, by explaining the implications of data non-normality on the interpretation of results or the choice of non-parametric statistical tests.
  4. It is recommended to enrich the "Results" section by introducing a "3.1 Experiments" subparagraph that incorporates the use of machine learning techniques, such as cluster analysis, to identify homogeneous groups of participants with similar characteristics. Clustering algorithms such as K-means, Hierarchical Clustering, or the Louvain method, which are well-known in the machine learning field, could be employed to group subjects based on key variables, providing a deeper understanding of how different subgroups respond to the intervention. This approach could enhance the interpretation of the data, offering a more detailed view of the study's results. For instance, it could help identify groups with specific demographic or clinical profiles that derive greater benefits from a particular type of intervention, thereby allowing for more targeted and personalized treatment strategies.
  5. It is suggested to separate lines 361 - 369 into a new paragraph titled "Implications, Limits, and Future Perspectives," enriching it and further implementing the existing analysis by supporting it with a broader review of the literature. This section should delve deeper into the implications of the results in relation to existing studies, highlighting how they might influence practice and theory in the field. Additionally, the study's limitations, such as sample size or methodological constraints, should be examined in greater detail, contextualizing them within the literature. Finally, future perspectives should be expanded by suggesting directions for further research and integrating references to relevant studies to provide a more comprehensive and informed framework.

Author Response

First of all, thank you very much for taking the time to read the article and for your support to make it better. The publication of this article is very important for my career. I would like to emphasize that I am always open to any kind of development with your valuable comments. I have explained the revisions I made based on your feedback in the word file I uploaded.

Reviewer 2 Report

Comments and Suggestions for Authors

Hello and thank you for choosing me as a reviewer

The topic is very practical and interesting and it deals with the effect of exercise on disabled people.

Demographic characteristics should be mentioned in the abstract.

The introduction is long. The importance and necessity of the work is not mentioned correctly

Complete demographic characteristics should be mentioned in sampling. What was the sampling method based on? Reference protocols should be mentioned.

As a result, it is better to use a figure to show the difference.

It is a long discussion and should be discussed mostly due to its effectiveness or lack of effectiveness.

Author Response

(The authors gave the same response as above.)

Round 2

Reviewer 2 Report

Comments and Suggestions for Authors

Article edited and its can be publish